# Elevated Lipoprotein(a) Level Influences Familial Hypercholesterolemia Diagnosis

**DOI:** 10.3390/diseases10010006

**Published:** 2022-01-18

**Authors:** Uliana V. Chubykina, Marat V. Ezhov, Olga I. Afanasieva, Elena A. Klesareva, Sergei N. Pokrovsky

**Affiliations:** 1A.L. Myasnikov Institute of Clinical Cardiology, National Medical Research Center of Cardiology, Ministry of Health of the Russian Federation, 121552 Moscow, Russia; uliankachubykina@gmail.com; 2Institute of Experimental Cardiology, National Medical Research Center of Cardiology, Ministry of Health of the Russian Federation, 121552 Moscow, Russia; afanasieva.cardio@yandex.ru (O.I.A.); hea@mail.ru (E.A.K.); dr.pokrovsky@mail.ru (S.N.P.)

**Keywords:** lipoprotein(a), familial hypercholesterolemia, Dutch Lipid Clinic Network criteria

## Abstract

Familial hypercholesterolemia (FH) and elevated lipoprotein(a) [Lp(a)] level are the most common inherited disorders of lipid metabolism. This study evaluated the impact of high Lp(a) level on accuracy Dutch Lipid Clinic Network (DLCN) criteria of heterozygous FH diagnosis. A group of 206 individuals not receiving lipid-lowering medication with low-density lipoprotein cholesterol (LDL-C) >4.9 mmol/L was chosen from the Russian FH Registry. LDL-C corrected for Lp(a)-cholesterol was calculated as LDL-C − 0.3 × Lp(a). DLCN criteria were applied before and after adjusting LDL-C concentration. Of the 206 patients with potential FH, a total of 34 subjects (17%) were reclassified to less severe FH diagnosis, 13 subjects of them (6%) were reclassified to “unlike” FH. In accordance with Receiver Operating Characteristic curve, Lp(a) level ≥40 mg/dL was associated with FH re-diagnosing with sensitivity of 63% and specificity of 78% (area under curve = 0.7, 95% CI 0.7–0.8, *p* < 0.001). The reclassification was mainly observed in FH patients with Lp(a) level above 40 mg/dL, i.e., 33 (51%) with reclassified DLCN criteria points and 22 (34%) with reclassified diagnosis, compared with 21 (15%) and 15 (11%), respectively, in patients with Lp(a) level less than 40 mg/dL. Thus, LDL-C corrected for Lp(a)-cholesterol should be considered in all FH patients with Lp(a) level above 40 mg/dL for recalculating points in accordance with DLCN criteria.

## 1. Introduction

Familial hypercholesterolemia (FH) and elevated lipoprotein(a) [Lp(a)] are the most common inherited disorders of lipid metabolism associated with an increased risk of cardiovascular disease (CVD). About 1.4 billion people worldwide have an Lp(a) level of more than 50 mg/dL [1], 30 million patients have FH [2], and at least 5 million have concomitant of FH and hyperlipoproteinemia(a) [hyperLp(a)] [3]. The most used clinical tool for the diagnosis of heterozygous FH is the Dutch Lipid Clinic Network (DLCN) criteria [4]. These guidelines consider genetic analysis, external manifestations of FH (tendon xanthomas and corneal arcus), personal and family history of early CVD, hypercholesterolemia in relatives, and untreated low-density lipoprotein cholesterol concentration (LDL-C). Depending on the presence of those or existing signs, the points obtained are summed up and the diagnosis of a definite, probable, or possible heterozygous FH is verified. Genetic testing is the gold standard for the diagnosis of FH, but with insufficient method availability [5], LDL-C plays the most important role in assessing the probability of FH: fewer points mean low LDL-C levels and more point denote high.

Estimations of LDL-C, direct LDL-C assays on automated chemistry analyzers, and even beta quantification after ultracentrifugation and lipoprotein separation include cholesterol from LDL and Lp(a) [6]. Thus, cholesterol in Lp(a) contributes to the reported concentrations of LDL-C in almost all currently available methods. In a person with a high level of Lp(a) expression, the cholesterol in this particle has the potential to contribute a significant portion of the reported LDL-C. Since the concentration of Lp(a) in patients with potential FH is 40–60% higher than in those with an unlikely FH, it is possible that Lp(a)-cholesterol contributes to an overestimation of the level of LDL-C in FH subjects [7]. This implies that some patients may have a clinical FH diagnosis, which is due to high Lp(a) concentration. Verification of the type of lipid metabolism disorder affects not only the stratification of the CVD risk, the choice of treatment tactics for these patients, but also in-fluences the FH clinical diagnosis. Information about the true LDL-C level can help physicians predict the effectiveness of cascade screening strategies and determine the need for FH genetic testing. We evaluated the impact of high Lp(a) level on the accuracy of DLCN criteria of heterozygous FH diagnosis.

## 2. Materials and Methods

The study included 206 FH patients from the Russian FH Registry, NCT02208869. (Figure 1). To clarify the probability of the FH presence, the DLCN criteria were applied. When making the diagnosis, we considered heredity for the early development of CVD and hypercholesterolemia in close relatives, history of the disease (presence and age of atherosclerotic CVD), data on the presence of tendon xanthomas and corneal arcus, genetic analysis of dyslipidemia (if possible) and the highest LDL-C level. Patients with nine or more points corresponded to definite FH diagnosis, from six to eight points were considered as probable, from three to five points as possible, and less than three points mean unlikely FH diagnosis.

LDL-C concentration was calculated by the Friedewald formula: LDL-C = TC − HDL-C − TG/2.2 (mmol/L). In addition, the level of corrected LDL-C (LDL-Ccorrected) was calculated, considering cholesterol of Lp(a): LDL-C corrected = LDL-C − 0.3 × Lp(a)/38.7 [8]. The concentration of Lp(a) was measured using an enzyme-linked immunosorbent assay using polyclonal antibodies to Lp(a), as previously described [9]. HyperLp(a) corresponded to Lp(a) level ≥30 mg/dL.

Genetic testing for FH was performed in 58 (28%) patients, in 38 (65%) of them pathogenic mutations were detected.

Statistical analysis was conducted with MedCalc 15.8 software (MedCalc Software Ltd., Ostend, Belgium). Qualitative variables were described by absolute numbers and percentages. Descriptive statistics of continuous variables after analysis of normality of distribution are presented as mean and standard deviation or median [25%; 75%]. The threshold value of Lp(a), its sensitivity and specificity were analysed by receiving curves of operational characteristics. Differences were considered statistically significant at *p* < 0.05.

## 3. Results

Table 1 presents the characteristics required for the establishment and stratification of the probability of FH diagnosis according to the DLCN criteria and Lp(a) level.

According to the DLCN criteria, 70 (34%) patients met the diagnosis of possible FH, 76 (37%) had probable and 60 (29%) had definite FH (Figure 2A).

To assess the effect of the Lp(a) level on the FH diagnosis, LDL-Ccorrected for Lp(a)-cholesterol level was calculated in 77 patients with hyperLp(a), the proportion of cholesterol in Lp(a) ranged from 2 to 74% (on average 12 ± 11%) of the LDL-C concentration of calculated using the Friedewald formula.

Recalculation of LDL-Ccorrected in patients with hyperLp(a) leads to an absolute change in LDL-C in the range from −0.24 to −1.59 mmol/L (median [25%, 75%]—0, 59 [0.34; 0.83] mmol/L) (Figure 3).

After using the LDL-Ccorrected, the DLCN criteria points were corrected. Figure 2B shows the distribution of patients depending on the FH probability after correction of LDL-C levels for Lp(a) concentration: unlikely in 13 (6%) patients, possible in 79 (38%), probable in 59 (29%), and definite in 55 (27%) subjects.

To test the predictive value of LDL-C level adjusted for Lp(a)-cholesterol for verification of the FH diagnosis according to the DLCN criteria we constructed a receiving curve of operational characteristics (Figure 4). Lp(a) level ≥40 mg/dL was associated with FH diagnosis reclassification with a sensitivity and specificity of 63% and 78%, respectively.

Of 206 patients with potential FH, 48 ones (23%) had points adjusted according to the DLCN criteria after recalculating the LDL-C concentration according to the Lp(a) level. A total of 34 subjects (17%) were reclassified to a less severe FH diagnosis, 13(6%) of them were reclassified to unlikely FH.

Table 2 shows the number of patients requiring reclassification according to the DLCN criteria depending on the Lp(a) level. Reclassification of the FH diagnosis was mainly observed in patients with Lp(a) levels ≥40 mg/dL. The proportion of individuals with reclassification of points after recalculating LDL-C was 51% (*n* = 33) and 15% (*n* = 21) in patients with Lp(a) ≥ 40 mg/dL and Lp(a) < 40 mg/dL, respectively. The proportion of individuals with reclassification of diagnosis according to the DLCN criteria was 34% (*n* = 22) and 11% (*n* = 15), respectively.

Thus, Lp(a) level ≥40 mg/dL has a significant impact on the accuracy of the DLCN criteria after recalculating LDL-C concentration.

## 4. Discussion

Currently, there has been growing awareness and alertness by physicians and the general population about the prevalence and pathogenicity of FH and elevated Lp(a) concentration. FH is a hereditary disorder of lipid metabolism characterized by elevated LDL-C level from birth [10]. The most frequently used diagnostic tool in the FH verification is the DLCN criteria, which consider a set of hypercholesterolemia signs and stratify them in the scoring [4]. The highest score of points is assigned a positive result of genetic testing and a high concentration of LDL-C. Genetic testing is the gold standard for the FH diagnosis, but its use is limited [5]. Since LDL-C and Lp(a) are apoB100-containing particles [11,12], Lp(a)-cholesterol is a component of global LDL-C level, measured directly or calculated using the Friedewald formula. The proportion of Lp(a) cholesterol can reach 30–45% of the total LDL-C level, especially in individuals with hyperLp(a) [13]. In our study, the proportion of Lp(a)-cholesterol from 2 to 74% of the LDL-C concentration calculated by the Friedewald formula in 77 patients with hyperLp(a). According to a recently published study [14], the average contribution of Lp(a)-cholesterol to LDL-C was 26% (range, 13–50%) for LDL-C measurements at LDL-C concentrations in the 10–39 mg/dL group. The percent contribution of Lp(a)-cholesterol to LDL-C decreased as LDL-C increased. In the highest four LDL-C groups, the average contribution of Lp(a)-C was 8.8% (range, 1–67%) of the total LDL-C measurement [14].

There have been a limited number of studies evaluating a high Lp(a) level impact for the FH diagnosis. The first study on this point published in 2016 was an analysis of the Copenhagen population (Copenhagen General Demographic Study), which showed that hyperLp(a) might be responsible for a quarter of previously diagnosed FH cases [7]. According to a recent Australian study, which included 907 patients with a potential FH diagnosis, 8.2% of patients according to the DLCN criteria were classified as unlikely FH after adjustment of LDL-C for Lp(a) level [15]. The proportion with probable and definite FH decreased significantly in patients with Lp(a) > 50 mg/dL (*p* < 0.01). In the US study [14] the proportion of FH patients defined by DLCN (definite, probable) criteria also decreased significantly after LDL-C concentration having been adjusted (*p* < 0.01). Of these patients, 36.1% (*n* = 119) were reclassified by DLCN. Of the 907 patients suspected of having FH, a total of 74 patients defined by DLCN criteria (8.2%) were reclassified to unlikely FH after adjusting LDL-C concentration for Lp(a) cholesterol [14]. In our study, in 23% of possible/probable/definite FH patients, points were adjusted after recalculating LDL-C for Lp(a) level: in 6% of patients, despite the correction of points, the diagnosis was not changed, but in 17% of subjects the diagnosis was reclassified to less severe (6% of them had the unlikely FH). Reclassification was mainly observed in patients with Lp(a) levels above 40 mg/dL (13% versus 35% determined by the DLCN criteria). Our data are consistent with others and indicate the importance of the Lp(a) concentration accounting when making the FH diagnosis. Thus, the use of a corrected LDL-C level for Lp(a)-cholesterol plays an important role in assessing the achievement of target LDL-C levels, especially in patients with hyperLp(a).

The accuracy of the FH verification and hyperLp(a) is of great importance not only in the stratification of CVD risk, but also in the potential effectiveness of lipid-lowering therapy and screening of relatives for FH. The strategy for the FH treatment is to aggressively reduce LDL-C and statins are the main class of lipid-lowering drugs in these patients, capable of reducing its level by 50% [5]. In some FH individuals LDL-C level after statin treatment are not sufficiently reduced to the target level. Insufficient decrease in LDL-C concentration may be partly due to Lp(a)-cholesterol contained in LDL-C in subjects with hyperLp(a). In general, statins do not significantly alter Lp(a) values (0 to + 7%) [16], while several studies describe a small decrease in Lp(a) by statins (−5%) [17,18]. Monoclonal antibodies to proprotein convertase subtilisin/kexin type 9 lead to a decrease in the LDL-C by 50–60% and are the only drugs used in clinical practice that diminish Lp(a) level by 30% [5,19]. The most effective currently available treatment for hyperLp(a) is lipoprotein apheresis [20].

Our study has some limitations. This is a cross-sectional sub-study of the Russian FH Registry with a small sample of subjects (*n* = 206), recruited and examined in the single center. Genetic testing was performed only in 28% of patients.

## 5. Conclusions

The results of our study show that the use of a corrected level of LDL-C based on Lp(a) concentration for FH clinical diagnosis according ti DLCN criteria leads to a significant diagnosis reclassification to less severe and reduces the likelihood of true FH. The LDL-C level corrected by Lp(a) cholesterol should be considered when making the FH diagnosis in patients with Lp(a) concentration ≥40 mg/dL. The proportion of cholesterol in Lp(a) can reach 74% of total LDL-C concentration in patients with hyperLp(a).

## Figures and Tables

**Figure 1 diseases-10-00006-f001:**
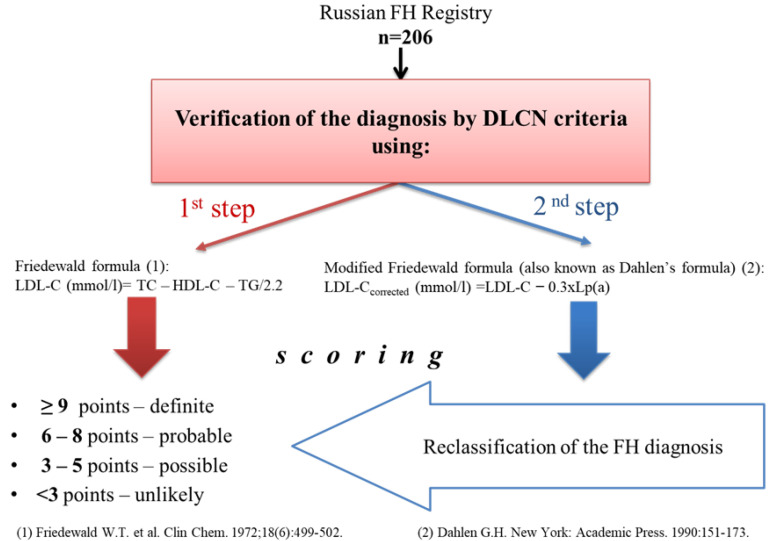
Study design.

**Figure 2 diseases-10-00006-f002:**
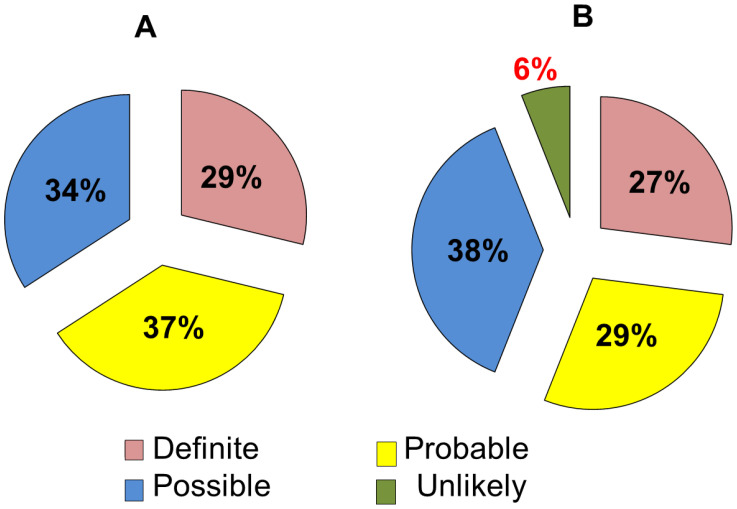
(**A**): Distribution of patients depending on the likelihood of familial hypercholesterolemia. (**B**): Distribution of patients depending on the probability of familial hypercholesterolemia after correction of low-density lipoprotein cholesterol level by lipoprotein(a). Note: Diagnosis of definite, probable, possible and unlikely familial hypercholesterolemia according to the Dutch Lipid Clinic Network criteria.

**Figure 3 diseases-10-00006-f003:**
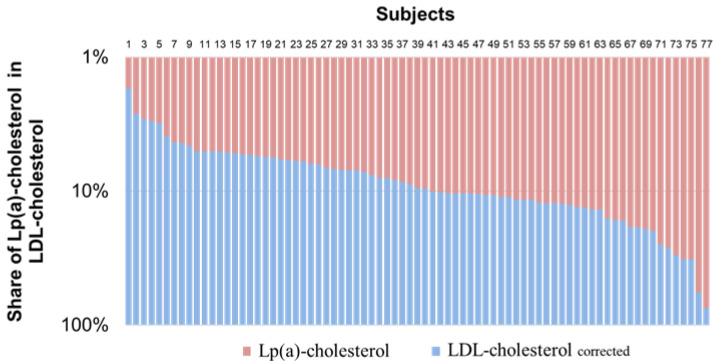
Contribution of lipoprotein(a) to the concentration of low-density lipoprotein cholesterol in patients with hyperlipoproteinemia(a) and familial hypercholesterolemia.

**Figure 4 diseases-10-00006-f004:**
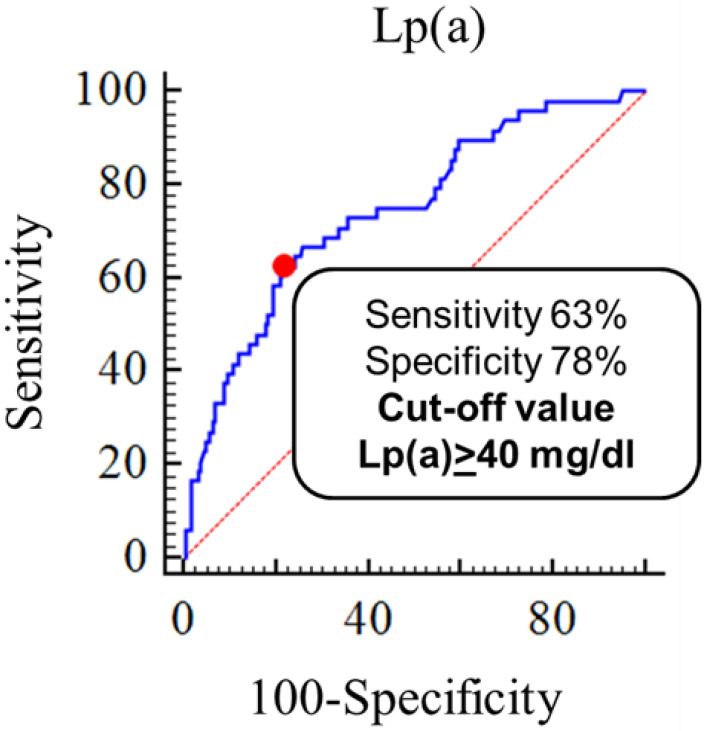
Threshold lipoprotein(a) level associated with heterozygous familial hypercholesterolemia re-diagnosing (AUC 0.7, 95% CI 0.7–0.8, *p* < 0.05).

**Table 1 diseases-10-00006-t001:** Familial hypercholesterolemia diagnosis according to the Dutch Lipid Clinic Network criteria and lipoprotein(a) level (*n* = 206).

Variable	Meaning
Relatives with LDL-C levels >95th percentile	153 (74%)
Family history of coronary heart disease	118 (57%)
Coronary heart disease	70 (34%)
Stroke/transient ischemic attack	11 (5%)
Genetic testing	58 (28%)
- *LDLR* mutation	35 (60%)
- *APOB* mutation	2 (3%)
- *PCSK9* mutation	1 (2%)
- no detected mutations	20 (35%)
Tendon xanthomas	25 (15%)
Corneal arcus	10 (5.5%)
Lipid profile	
Highest LDL-C level, mmol/L	6.5 ± 2.4
Lipoprotein(a), mg/dL	17 [7;52]
LDL-Ccorrected, mmol/L	6.2 ± 2.4
Lipoprotein(a) ≥ 30 mg/dL	77 (37%)

Note: LDL-C—low density lipoprotein cholesterol, LDL-Ccorrected—low-density lipoprotein cholesterol corrected for cholesterol included in lipoprotein(a), LDLR—low-density lipoprotein receptor gene, APOB—apolipoprotein B gene, PCSK9—proprotein convertase gene subtilisin/keksin type 9 gene.

**Table 2 diseases-10-00006-t002:** Effect of Lp(a) concentration ≥40 mg/dL on reclassification FH diagnosis (*n* = 206).

FH Diagnosis			Unlikely	Possible	Probable	Definite
**Lp(a) < 40 mg/dL**	*n* = 141	Diagnosis	0	49	56	36
		Diagnosis _corrected_	5	52	49	35
		Reclassification of subjects	+5	+3	−7	−1
		Total % of reclassified points	**15%**			
		Total % of reclassified diagnosis	**11%**			
**Lp(a) ≥ 40 mg/dL**	*n* = 65	Diagnosis	0	21	20	24
		Diagnosis _corrected_	8	25	10	22
		Reclassification of subjects	+8	+4	−10	−2
		Total % of reclassified points	**51%**			
		Total % of reclassified diagnosis	**34%**			

Note: FH—familial hypercholesterolemia, Lp(a)—lipoprotein(a), Diagnosis corrected—DLCN criteria for the diagnosis of FH, considering the corrected level of LDL-C for cholesterol included in Lp(a).

## Data Availability

The data presented in this study are available on request from the corresponding author. The data are not publicly available due to privacy restrictions.

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
