# Peer review of "Elevated Lipoprotein(a) Level Influences Familial Hypercholesterolemia Diagnosis"

_diseases, 2022, doi:10.3390/diseases10010006_

Round 1
Reviewer 1 Report
After critically reviewing this Research Article titled "Elevated lipoprotein(a) level influences on familial hypercholesterolemia diagnosing", I detected some MINOR flaws, which determined my recommendation of “ACCEPT UNDER REVIEW”. Below please find my detailed comments.
The authors studied the impact of high lipoprotein(a) [Lp(a)] level on accuracy Dutch Lipid Clinic Network (DLCN) criteria of heterozygous Familial hypercholesterolemia (FH) diagnosing, and changes in this diagnosis could lead to a reclassification of patients and, consequently, a change in the therapeutic management of the disease..
The study is well written and conducted, with well-described methodology, with minor flaws and easily corrected and appropriate for the objectives of the work. Statistical analyzes of the results obtained were well conducted.
The results obtained were very promising and the discussions were relevant to the results achieved.
- If it is still possible, I suggest that the research be registered with some government clinical studies agency, such as ClinicalTrials.gov.
- The authors do not describe how the classification of patients with diagnosis of heterozygous FH is made, whether there is any genetic test for this, or is it just observational.
The limitations of the study were not well pointed out and should be written in the end of the article.
Reviewer 2 Report
Interesting study not novel- see ref-3
the sample is small/
however the conception described quite well
I would expect to provide for the readers description of clinical features of the patients (XANTOMAS, CAD etc... and points for diagnosis
